# The hippocampus dissociates present from past and future goals

Alison Montagrin [1,2,3,9] ✉, Denise E. Croote [1,9], Maria Giulia Preti [4,5,6], Liron Lerman[7], Mark G. Baxter [1] & Daniela Schiller [1,8] ✉

Our brain adeptly navigates goals across time frames, distinguishing between urgent needs and those of the past or future. The hippocampus is a region known for supporting mental time travel and organizing information along its longitudinal axis, transitioning from detailed posterior representations to generalized anterior ones. This study investigates the role of the hippocampus in distinguishing goals over time: whether the hippocampus encodes time regardless of detail or abstraction, and whether the hippocampus preferentially activates its anterior region for temporally distant goals (past and future) and its posterior region for immediate goals. We use a space-themed experiment with 7T functional MRI on 31 participants to examine how the hippocampus encodes the temporal distance of goals. During a simulated Mars mission, we find that the hippocampus tracks goals solely by temporal proximity. We show that past and future goals activate the left anterior hippocampus, while current goals engage the left posterior hippocampus. This suggests that the hippocampus maps goals using timestamps, extending its long axis system to include temporal goal organization.

Tracking personal goals is a vital and ongoing cognitive process. Our brains are continuously monitoring what we have already accomplished, are currently pursuing, and are planning to tackle in the future. For instance, we know that we no longer need to go to the grocery store, but we do need to pay the credit card bill today to prevent accruing interest. These goals differ on a fundamental parameter: time. While some goals must be accomplished in the present, others are removed in time from the needs of the current self, representing goals that were either completed in the past or remain to be accomplished in the future. Yet, goals are dynamic entities. Take, for instance, the act of mailing a wedding gift – it starts as a future goal, evolves into a current priority, and then to a completed goal as we move forward in time. Therefore, effectively tracking personal goals involves managing different goals and updating the relevancies of each goal in memory as time progresses.

Here, we investigated how the brain keeps an updated representation of achieved goals, goals demanding immediate attention, and goals that will be relevant later in time, using functional magnetic resonance imaging (fMRI). We focused our investigation on the hippocampus, due to its established role in episodic memory and mental time travel[1–3].

Substantial literature has identified anatomical and functional specialization along the hippocampal longitudinal axis, notably in the episodic and spatial domains, with increasing representational granularity appearing more posteriorly along the long axis[4–6]. Observations regarding place cells in rodents have unveiled a fascinating nuance: in consistent locations, these cells seem to represent time or moments

[1]The Nash Family Department of Neuroscience, Icahn School of Medicine at Mount Sinai, New York, NY 10029, USA. [2]Department of Neuroscience, University of Geneva, Geneva 1202, Switzerland. [3]Swiss Center for Affective Sciences (CISA), University of Geneva, 1202 Geneva, Switzerland. [4]CIBM Center for Biomedical Imaging, Lausanne, Switzerland. [5]Neuro-X Institute, École Polytechnique Fédérale de Lausanne (EPFL), Lausanne, Switzerland. [6]Department of Radiology and Medical Informatics, University of Geneva (UNIGE), Geneva, Switzerland. [7]Sector 5 Digital, New York, NY 10018, USA. [8]Department of Psychiatry, Icahn School of Medicine at Mount Sinai, New York, NY 10029, USA. [9]These authors contributed equally: Alison Montagrin, Denise E. Croote. ✉e-mail: alison.montagrin@unige.ch; daniela.schiller@mssm.edu

within sequences, separate from spatial positions[6–9]. It has been suggested that such 'time fields' expand from dorsal to ventral regions in a manner comparable to spatial place fields[6]. This prompted us to ask whether information mapping along the hippocampal long axis extends to the temporal domain: are goals mapped along the long-axis based upon their temporal distance from the present?

Examining the impact of temporal distance on goal representations, however, poses a notable methodological challenge. Comparing representations of items close in time to those distant in time, whether they are past memories or future projections, introduces inherent differences in the content, realness, and level of detail associated with these constructions[10–12]. These experiential disparities can then cloud the interpretation of both behavioural and neural time-based observations. To address this limitation, we designed a paradigm that isolates the temporal elements of goals. We held the properties of the goals fixed (i.e., their names and visual features), and the level of detail associated with present, past, future, near, and distant goals constant. The sole parameter that we changed about a goal throughout our experiment was its temporal distance from the present.

To accomplish this, we framed goals in the context of an imaginary space mission, rather than relying on goals derived from participants' personal pasts and futures. Participants embarked on a 4-year mission to Mars, necessitating the completion of a series of goals crucial for their survival on the planet. These goals varied in when they needed to be accomplished during the mission, with different sets of goals applicable in the first, second, third, and fourth year of the journey. As participants advanced through the mission, the *same* goals were presented to them; however, the participants themselves moved forward in time, which led to a shift in the temporal relevance of the goals. Goals initially designated as future-oriented transitioned into current needs, while the current needs transformed into goals of the past. Thus, participants had to manage multiple goals that differed in their temporal distance and update their goal representations as they moved through their 4-year mission on Mars. We hypothesized that current goals would trigger posterior activation in the hippocampus, given their temporal proximity to the present, while goals temporally removed in time would trigger anterior activation in the hippocampus, due to their temporal distance from the present. To focus our analyses on the hippocampal long axis, we combined multi-echo multi-band imaging with ultra-high field 7T fMRI techniques[13,14]. We found that goals pertaining to the past and future activated the left anterior hippocampus, whereas goals pertaining to the present activated the left posterior hippocampus. These findings demonstrate that the brain evaluates goals differently depending on their temporal relevance, distinguishing between those that are current and those that are temporally remote.

## Results

A sample of 34 neurologically healthy individuals completed the two-day experiment and 31 remained in the sample after MRI exclusions ($M = 27.0$ years, $SD = 4.5$, Range = 19–36, $n = 15$ females; see Methods). Participants were taught when they needed to complete each goal via a training exercise on the first day of the experiment (Fig. 1a). In addition to learning goals relevant for the first, second, third, and fourth years of the mission, participants also learned a set of goals that they always needed to accomplish and a set of goals that they never needed to accomplish. Participants learned the relevancy of five goals for each of the 6 aforementioned timeframes, totalling to 30 stimuli (see Methods for training details).

On the second day of the experiment, participants were trained on the paradigm in a mock scanner and then sent on their mission in an ultra-high field 7T MRI scanner (Fig. 1a). They were shown a goal and asked when they need to accomplish it, with buttons present to indicate whether the goal was applicable for the current Mars' year, or relevant in the distant past, near past, near future, distant future,

always, or never (Fig. 1b). Distant goals were defined as those 2-3 years from the present and near goals within 1 year of the present during the mock scanner training session. Participants evaluated when they needed to accomplish each of the 30 goals before advancing to the next year of the mission. They were then shown the *same* 30 goals, but the goals' temporal relevancies shifted now that the participants had moved forward in time (Fig. 1c). For example, when participants were in their 2nd year of the expedition and were presented with a goal relevant in the 3rd year, they selected the near future button (NF) on the screen to indicate that this goal should be attended to in the near future. When they advanced to the 3rd year of the game, this goal shifted to a current priority and participants selected the current year button (CY). Therefore, participants had to continuously apply their knowledge of the timing of the goals and imagine when they needed to complete each goal in relation to their own temporal position (see Methods for a full task description).

As the mission progressed, new buttons appeared on the screen allowing participants to indicate whether they had already completed the goal displayed in the past, and buttons were removed from the screen as the future timeframe disappeared throughout the mission. The trials were divided into instances where participants were making an evaluation of a goal that needed to be completed in the distant future (DF), near future (NF), near past (NP), and distant past (DP; 15 trials each), or in the current year (CY), always needed (AN), and never needed (NN; 20 trials each), totalling to 120 trials. All goals revolved around space shuttle maintenance, space suit care, personal nutrition, exercise, and recreational activities (Fig. 1d).

## Behavioural results

We first explored the behavioural impact of the timestamps attached to the goals. We compared reactions times for goals that were removed in time (distant future, near future, near past, distant past) to reaction times for current goals. If participants are using a mental time travel-like process to evaluate the goals, we expected longer response times for the temporally removed goals in the past and future compared to the goals in the present year.

We examined participants' reaction times using a series of linear mixed models, implemented via the lme4[15] and lmerTest[16] packages in R v. 3.6.0[17]. After removing incorrect and reaction time outlier trials (+/− 3 SD's of a participant's mean) 94.5% of all trials remained. We modelled participants' log transformed reaction times (this transformation aimed to address skewness and stabilize variance across conditions, making the data more suitable for parametric analysis) against temporal condition, with levels for distant future, near future, current, near past, and distant past trials. Current trials were distributed equally throughout the game, while future trials were presented in the beginning 75% and past trials in the last 75% of the mission. This distribution was in effect because there were no future goals in the last year of the mission, and no goals had already been completed in the first year of the mission (see Supplementary Table 2 for a visualization of the trial numbers).

We included a random effect for participant ID to account for the lack of independence between trials within a participant, and game year was included as a fixed effect and as a random effect for objective time to account for any trends in reaction time within and between participants across the years as the mission proceeded (Table 1). This game year regressor has four levels (year 1, year 2, year 3, year 4) and was in place to absorb variance such that any reaction time differences that remained between the conditions were those above and beyond what could be explained by time point in the game.

We evaluated statistical significance using Type II Sums of Squares and computed the model-based estimated marginal means, standard errors, and confidence limits via the emmeans package in R. All post hoc contrasts quantified the difference between the two estimated marginal means and outputted the corresponding estimate, standard

## a     Experimental procedure

**Day 1** — Training session (~1.5 hr) — **Day 2** — Mock scanning session (~30m) — 7T experimental scanning session (~1 hr) — Post task debrief (~15m)

## b     7T experimental scanning session

## Example question

When do you need to accomplish this goal?

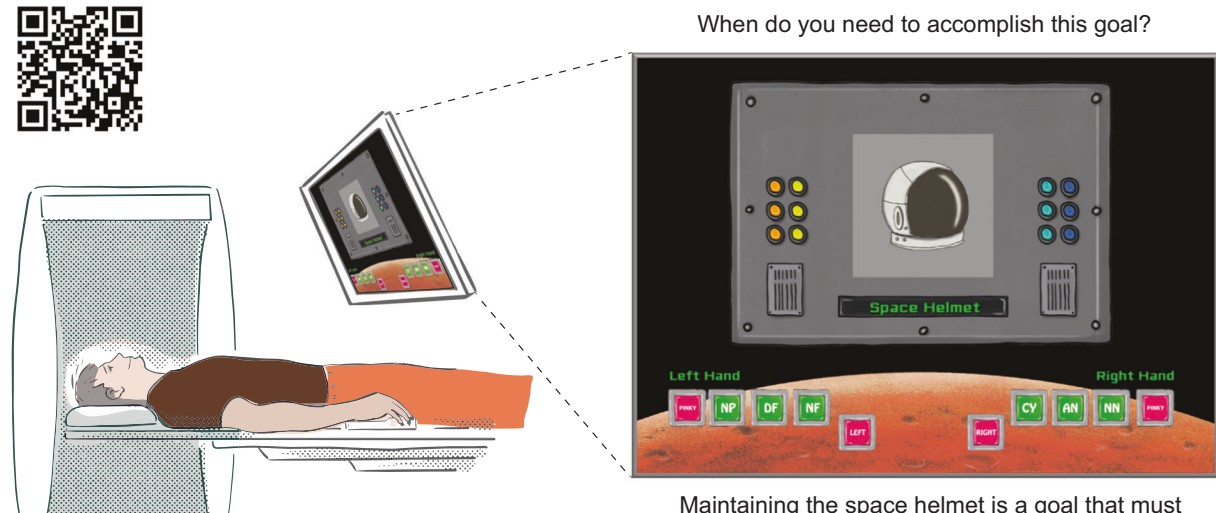

Maintaining the space helmet is a goal that must be accomplished in the **3rd** Mars year.

## c     Space helmet goal trajectory

**YEAR 1**

Goal is relevant in the distant future.

**YEAR 2**

Goal is relevant in the near future.

**YEAR 3**

Goal must be completed in the current year.

**YEAR 4**

Goal was accomplished in the near past.

**Mission complete**

## d     Example set of goals

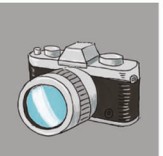
take scenic photographs

space helmet

eggs

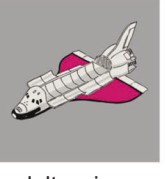
delta wings

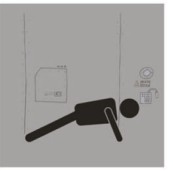
push ups

**Fig. 1 | Experimental design. a** The experiment took place over two days. Participants learned when they would need to complete each goal during a training session on the first day of the experiment. Participants were familiarized with the paradigm in the mock scanner and then embarked on their mission inside the 7T scanner on the second day. **b** Example screen shown during year 1. While lying in the scanner, participants were shown a goal and asked when they need to accomplish this goal, with buttons at the bottom of the screen to indicate whether the goal was applicable for the current Mars year (CY), or relevant in the distant future (DF), near future (NF), near past (NP), distant past (DP), always (AN, always needed), or never (NN, never needed). The QR code grants access to the video shown to the participant at the beginning of the experiment to actively involve participants in the task by immersing them in an engaging experience, depicting a rocket's journey from Earth, through space, and landing on Mars. **c** Example trajectory of the space helmet goal across the mission, which is relevant in the 3rd Mars year for participants completing version 1. **d** Example set of goals. Participants learned 6 sets of 5 goals, each set associated with either year 1, year 2, year 3, year 4, always, or never.

**Table 1 | Reaction time linear mixed models**

| Model | Description | Formula |
|---|---|---|
| 1 | Relationship between participants' log transformed reaction times and temporal condition | Log RT~ game year + temporal condition + (1 + game year | participant ID) |
| 2 | Relationship between participants' log transformed reaction times and trial type | Log RT~ game year + trial type + (1 + game year | participant ID) |

The dependent variable was comprised of participant's log transformed reaction times for each trial. The temporal condition predictor in **Model 1** included five levels, distant future, near future, near past, distant past, and current trials. Always and never trials were filtered from the dataset for Model 1. The trial type predictor in **Model 2** included two levels, trials with a temporal element (distant future, near future, near past, distant past trials) and trials without (current, always, never trials). The participant ID random effect contained 31 participants and was included to account for the lack of independence between trials within a participant. The game year fixed effect and random effect included 4 levels (year 1, year 2, year 3, year 4) and served to account for downward reaction time trends as the game progressed between and within participants.

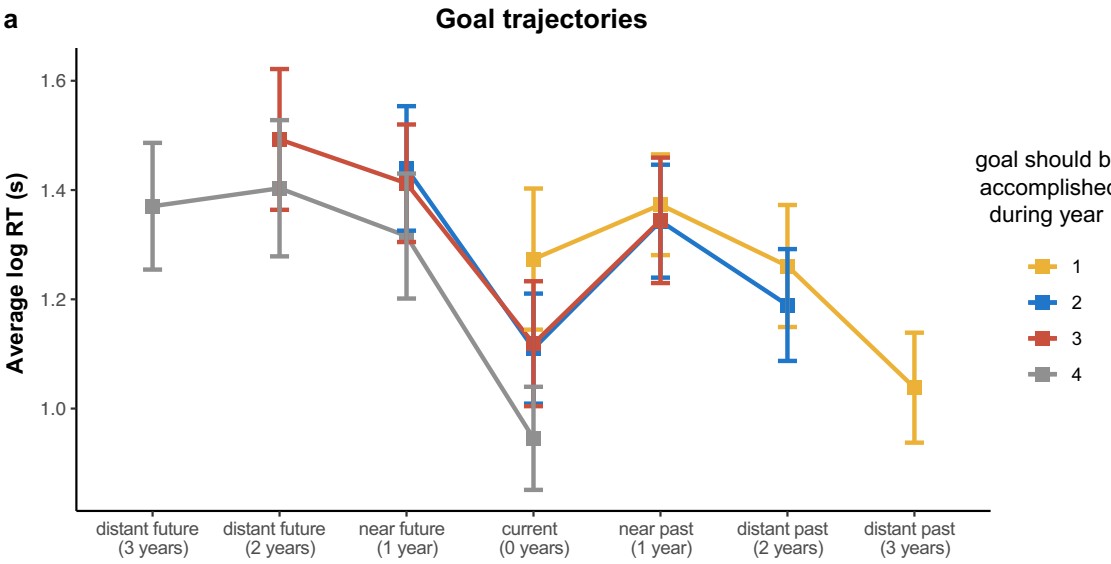

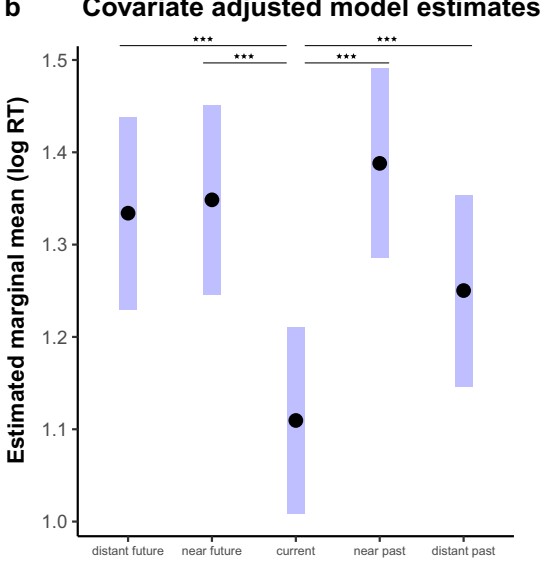

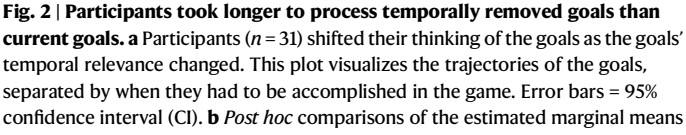

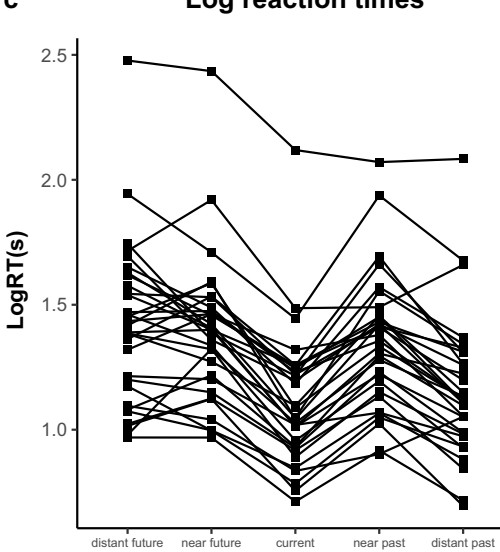

**Fig. 2 | Participants took longer to process temporally removed goals than current goals. a** Participants ($n = 31$) shifted their thinking of the goals as the goals' temporal relevance changed. This plot visualizes the trajectories of the goals, separated by when they had to be accomplished in the game. Error bars = 95% confidence interval (CI). **b** *Post hoc* comparisons of the estimated marginal means revealed that current goals were processed faster than all temporally removed goals. ***$p < 0.001$. (Supplementary Table 3 provides a comprehensive overview of all contrasts). Error bars = 95% CI. **c** Individual participants' average log reaction time for each of the temporal conditions.

error, test statistic, and *p* value[18]. Post hoc tests were corrected for multiple comparisons using a Tukey adjustment.

The reaction times to process the same goals shifted based upon the trajectories of the goals, i.e., how their temporal relevance changed across the game (Fig. 2a). This suggests that participants were indeed reframing how they thought about each goal as its temporal context shifted. Participants' log-transformed reaction times were significantly associated with temporal condition (Type II ANOVA; $F(4,$

2236.3) = 64.83, $p < 0.001$). Post hoc comparisons revealed that, as hypothesized, current trials were processed faster than all temporally removed conditions when contrasted individually (Fig. 2b, c; Supplementary Table 3).

We validated the findings above using a second model that also included the always and never trials. Using this model, we compared between trials with and without temporal computation. As with the current goals, always and never goals did not include a temporal element. We separated trials into those involving a temporal computation (distant future, near future, near past, distant past) and those not (current, always, never) and likewise found that participants took significantly longer to process trials involving a temporal element (Supplementary Fig. 1). We then examined the trials without temporal computation more closely. We introduced this control analysis as a response to alternative explanations associated with response times (RTs) or task demands. Specifically, we considered the 'without temporal computation' condition, encompassing current, always, and never needed conditions. Post hoc comparisons revealed that participants were faster to process goals that are irrelevant (never needed) than goals that are relevant at one point (current) or always needed. Always and current trials did not differ significantly (Supplementary Fig. 2 and Table 1). In summary, our behavioural findings provide evidence that the brain evaluates goals relevant to one's current situation differently than it does those that are removed in time.

In an additional type of trials, participants had to choose between two goals: one relevant for the current year and one relevant for another timeframe. We used a similar linear mixed model as above to examine the additional choice trials (see Supplementary Fig. 3). Post hoc comparisons of the estimated marginal means revealed that participants' choices between current and distant future goals were faster than choices between current and near future goals, and current and near past goals. These results are possibly consistent with the pattern separation theory (e.g., ref. 19). The boundary between the goals in the current moment and those that are near in time is more blurred compared to goals that are further away in time where the boundary with the present would be sharper and would take less time to distinguish.

### Neuroimaging results

In our main general temporal hypothesis, we sought to identify regions within the hippocampus that showed differential activity for temporally removed and current goals using General linear modelling analyses (GLM). We defined a series of general linear models (GLMs) with regressors for each temporal condition, the always and never trials, and years 1 – 4 of the game (see Methods for full model descriptions and Supplementary Tables 4 and 5). We first contrasted blood-oxygen-level-dependent (BOLD) activity for the grouped temporally removed conditions (distant future, near future, distant past and near past) with that of the current condition, and vice versa, bilaterally in the hippocampus (see Supplementary Fig. 4 for the parameter estimates). We used an automated subcortical segmentation methods FreeSurfer v7.2 to measure volumes of these regions of interest (i.e., bilateral hippocampus, see Methods for more details).

We hypothesized that signal amplitude would differ along the long axis as a function of temporal distance. Consistent with this hypothesis, temporally removed goals (remote condition = distant future + near future + distant past + near past) produced stronger activation than current goals (current condition) in the left anterior hippocampus (Fig. 3a; Table 2). The contrast remote > current identified voxels falling anterior to Y = −21 (mm in MNI coordinate). Current goals produced stronger activation in the left posterior hippocampus, with voxels extending from Y = −31 to Y = −36 (Fig. 3b; Table 2). These voxel locations align with the posterior and anterior boundaries outlined in ref. 4, who defined the foci of the anterior hippocampus to y = −21, using the uncal apex as the anatomic landmark. The same goals were anatomically dissociated along the longitudinal axis based on whether they were currently relevant, or relevant at a point removed in time (Fig. 3c, d; see Supplementary Table 6 for temporal gradient analysis). Although we observed a dissociation across the long axis in the left hippocampus, activation differences were not as prominent in the right hippocampus (Table 2). As post hoc analyses, we examined conditions that did not involve temporal computation but were distinct from the current condition. Specifically, we focused on the always and never needed conditions, which were associated with goals that were always needed to be accomplished or never needed to be accomplished, respectively (Supplementary Table 1, Supplementary Fig. 2).

All analyses were corrected for multiple comparisons implementing Family-Wise Error (FWE) using GRF-theory based maximum thresholding (voxel-wise correction, two-tailed $p = 0.025$).

Though our a priori hypotheses were centred on the hippocampus, we further conducted an exploratory whole brain analysis. This GLM was designed as described above. All statistical maps were masked with a grey matter tissue probability map and corrected for multiple comparisons using voxel-wise correction thresholding (FWE voxel-wise correction, two-tailed $p = 0.025$). Temporally removed goals largely activated anterior brain region, whereas current goals more strongly activated posterior brain regions (Supplementary Fig. 5, Supplementary Table 7). This separation, namely the localization of temporally removed goals to anterior brain regions, is in line with the role of the frontal cortex in autobiographical planning and future mental time travel[20,21].

We extended our investigation for the general temporal hypothesis by integrating a General Psychophysiological Interaction (gPPI) analysis. We explored how the anterior and the posterior hippocampus, serving as seed regions, engage in dynamic communication with other brain regions in response to goals that vary in temporal distance (see Supplementary Fig. 6, Supplementary Note 1, Supplementary Tables 8 and 9).

In summary, the left hippocampus anatomically distinguished present goals from goals that were removed in time along its longitudinal axis. Temporally removed past and future goals activated the anterior hippocampus, while current goals activated a more posterior portion of the hippocampus. We observed this temporal dissociation to a lesser degree in the right hippocampus, which similarly showed stronger anterior activation for temporally removed past and future goals.

## Discussion

In this study we investigated whether goals are mapped along the hippocampal anterior-posterior axis based on when they need to be accomplished. We examined how participants represented goals that were a current priority, goals that they accomplished in the past, and goals that they needed to complete in the future using a space mission themed paradigm. We designed this paradigm such that all properties of the goals were fixed, except their temporal relevancies, which updated as participants progressed throughout their mission. By rotating the temporal distances of identical stimuli, we sought to ensure that our findings were directly associated to the temporal distance of the goals. This approach effectively ruled out any potential influence stemming from inherent differences in experiential properties among past, future, close, or distant entities.

Behaviourally, goals that were relevant for ones' current needs were processed faster than temporally removed past and future goals. On a neural level, current goals activated the left posterior hippocampus and temporally removed past and future goals activated the left anterior hippocampus. Altogether, these findings extend the known scope of the hippocampus's long-axis system to the goal temporal-mapping domain. They demonstrate that the mental timestamps assigned to goals that are otherwise identical, guide their dissociation along the anterior-posterior parts of the hippocampus.

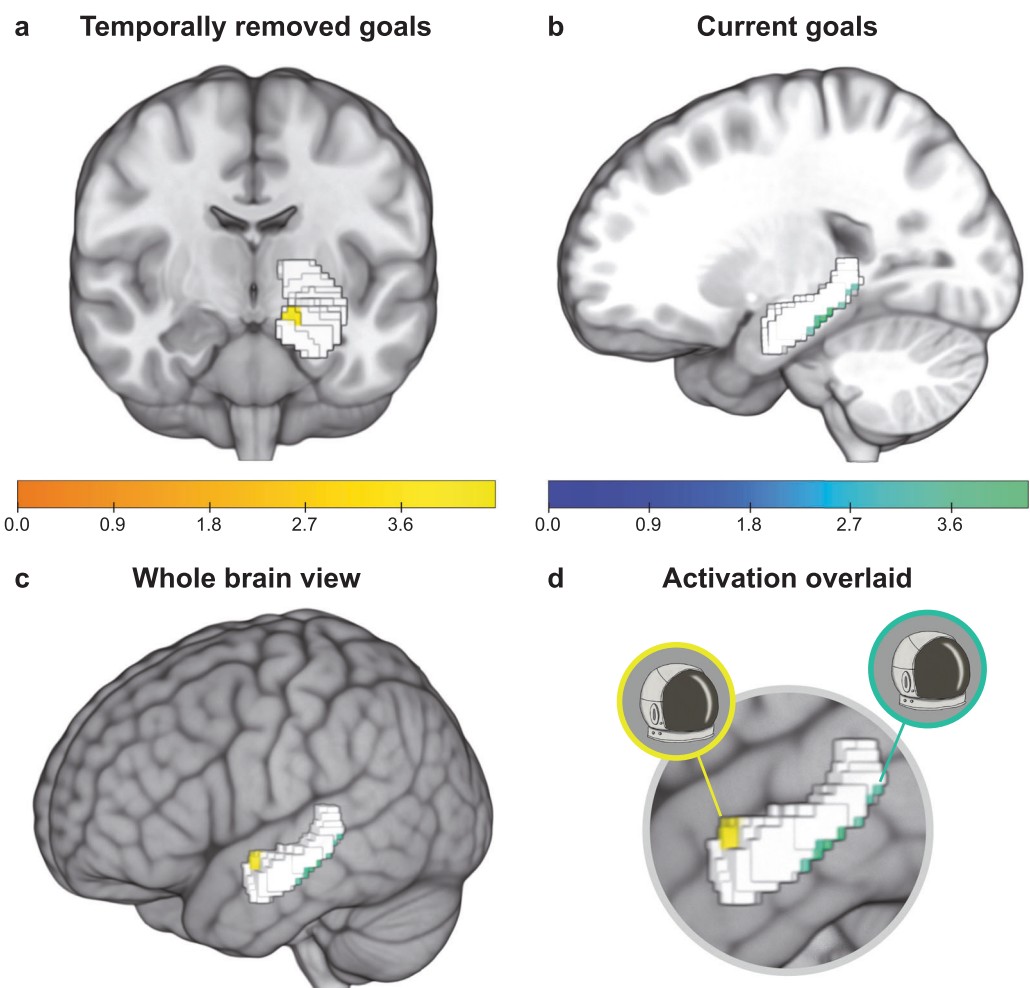

**Fig. 3 | Temporally removed goals activated the left anterior hippocampus and current goals activated the left posterior hippocampus. a** Activation maps for the contrasts comparing the remote (distant future + near future + distant past + near past) > current are overlaid in yellow. **b** Activation maps for the contrasts comparing the current > remote are overlaid in green. All z-statistic images were thresholded parametrically using GRF-theory-based maximum height thresholding with a (FWE-corrected) significance threshold of *p* = 0.025. **c** Activation for the temporally removed goals (yellow) and the current goals (green) shown concurrently on the brain. **d** The same goal, for instance fixing the space helmet, was anatomically dissociated along the longitudinal axis based on whether it was currently relevant, or relevant at a point removed in time. Contrast maps were overlaid and rendered onto a 3-dimensional MNI 152 brain using MRIcroGL. The left hippocampal region of interest (ROI) is displayed in white. Color bars reflect the thresholded z-statistic scores.

Long axis specialization has been explored in the episodic memory and spatial navigation domains. This works has largely focused on differences in representational granularity along the longitudinal axis and has found evidence of a detail-rich to gist-like gradient system[4–6]. Small scale event information[22] and detailed spatial information[23] are represented in the posterior hippocampus, while multi-level narrative information[22] and contextual spatial information[23] activate the anterior hippocampus. These functional divisions are supported by anatomical dissociations, where larger posterior and smaller anterior hippocampal volumes predict navigational proficiency in taxi drivers[24,25] and superior recollection memory[26]. We extend this literature by demonstrating that representations of personal goals are also dissociated along the axis, in this context based upon their temporal distance from the present.

Examining mental constructions over time, be it past memories or future simulations, is methodologically challenging. Changes in temporal distance are inherently accompanied by changes in the properties of the representations. Construal level theory posits that humans form more abstract construals of distal entities than they do of proximal entities, where construal level refers to the degree of concreteness or abstractness of the mental representation[11,12,27,28]. High-level construals are abstract, coherent, and schematic representations and are associated with distal objects, actions, or contexts. Low level construals contain concrete, idiosyncratic, and incidental information and are assigned to proximal entities[12,28]. As a result, it becomes difficult to disentangle neural and behavioural effects arising from increases in temporal distance from those induced by changes in the nature of the constructions. We experimentally removed differences in construal level by holding the content, realness, and level of detail associated with our goals constant. We selectively varied the temporal distance of the goals, and our results demonstrate that the temporal distance alone is sufficient to elicit hippocampal long axis dissociations. It is likely that when we organize our goals in the past or future, having the goals in mind in an abstract way is sufficient, whereas when we are in the present, we need more details of the goal to be able to achieve that goal.

Cortical and subcortical regions outside the hippocampus are also heavily involved in goal cognition. For instance, the ventromedial

**Table 2 | Coordinates of peak activation for temporally removed and current goals in the left and right hippocampus**

| Contrast | Voxel size | Hemisphere | Axis Hippocampus | Z-stat | Coordinates in mm (MNI) | | |
|---|---|---|---|---|---|---|---|
| Current > Remote | 8 | L | Posterior | 4.28 | −30 | −31 | −14.5 |
| | 1 | L | Posterior | 3.75 | −30 | −36 | −12 |
| Remote > Current | 8 | L | Anterior | 4.48 | −15 | −8.5 | −14.5 |
| | 1 | R | Anterior | 3.79 | 17.5 | −21 | −14.5 |

Temporally removed (past and future) goals activated the left anterior hippocampus, while current goals activated mainly the left posterior hippocampus, and one voxel in the right hippocampus. This table reports the number of significant voxels in the cluster, maximum z-statistic within the cluster, and the x, y, z location of the maximum intensity voxel for each contrast. Coordinates are reported in Montreal Neurological Institute (MNI) space. All statistical maps were corrected for multiple comparisons using maximum height thresholding (FWE voxel-wise correction, $p = 0.025$).

prefrontal cortex, including the medial orbitofrontal cortex and ventral medial cortex, represents the relative values of different goals[29,30]. The ventral striatum processes situations that are goal relevant for individuals[31] and computes prediction errors, i.e., deviations from reward expectations, which service goal directed behaviours by supplying updated value estimates of the world[32,33]. These bodies of work have examined how the brain encodes the values of different goals, ranks possible outcomes, and selects actions that will lead to goal attainment. We distinguish ourselves from these studies by examining how the brain represents the timestamp associated with the same goals as the goals evolve though the past, present, and future.

We found that current goals were processed more quickly than temporally removed goals. We theorize that this dissociation reflects the preferential status given to current needs over those that are removed, and the additional time required to mentally travel in time to place the past and future goals on a timeline. This observation is consistent with reaction time distance effects observed in temporal judgment tasks, where participants take longer to make judgments from a past or future timepoint versus a present timepoint[34,35]. Further, current goals activated more posteriorly and temporally removed goals more anteriorly in the left hippocampus. These results could potentially be attributed to the inherent time-related computations required for goals placed on a timeline. Regarding conditions without temporal constraints, goals categorized as never relevant exhibited swifter processing compared to goals designated as current or always relevant. This observation highlights a possible mechanism that governs the adjustment of goal processing based on temporal context (when goals matter) and relevancy (do goals matter).

We speculate that the timestamp of past and future goals may be localized in the anterior hippocampus to be integrated into schemas characterizing the time periods when the goals were already accomplished or must be accomplished[28]. Poppenk, et al. [4] proposed that the anterior hippocampus plays a role in locating these episodic memories, while the detailed content of such memories is retrieved from the posterior hippocampus. Projecting into the future, according to Conway et al.[36–38], does not necessarily demand access to the intricate details of the episodic memory (goal); an abstract representation suffices. Similarly, for a past goal, there is no need for specific details, its general representation is sufficient to remember the status of the goal (e.g., achieved).

Furthermore, the findings of remote (anterior) versus current (posterior) representation of goals in the hippocampus in our study are in line with animal studies showing different spatial scales along the longitudinal axis of the hippocampus. Specifically, the anterior (ventral) sections seem to 'zoom out', capturing broader spatial extents greater than 10 meters, whereas the posterior (dorsal) regions tend to 'zoom in', focusing on more localized spatial scales of under 1 meter with great distances (>10 m)[4,6,39]. Thus, it is possible that an analogue memory temporal structure is represented in the human brain. However, more studies are needed to confirm these hypotheses.

The anterior hippocampus has preferential access to the brain's motivational and schema circuitry through connections with the amygdala, nucleus accumbens, insula, vmPFC, perirhinal cortex, and ventral tegmental area[40–43]. The anatomical distinction observed may

reflect the need for past and future goals to be placed on a personal timeline and monitored in relation to other episodic and autobiographical events in one's life[44]. Finally, Thorp et al. [45] suggested that taking into account the anterior medial and lateral parts of the hippocampus might bring new insights into a possibly non-linear representation of the granularity along its axis.

We found that goals were mapped based on their temporal distance more robustly in the left hippocampus than in the right hippocampus. We had no a priori assumptions about left and right hippocampal lateralization, but our findings align with previous work citing lateralized hippocampal activation. The right hippocampus has been more heavily involved in allocentric spatial memory and navigation[46–50] and the left hippocampus in egocentric associative and sequential memory[49–54]. Participants in our task were tracking goals that needed to be sequentially accomplished and were processing the goals in relation to their present selves, thus envisioning the goals from an egocentric point of view. Our hippocampal lateralization contributes to the notion that the two hippocampi provide complementary representations of the world, rather operating via a single unified function[49].

A limitation of this research is that participants could have memorized the stimuli and relied on mnemonics to answer the questions, rather than incorporating time into their evaluations of the goals. We thoroughly trained the participants on the mission framework to reduce this risk, and we heavily emphasized during the training that it was important for them to think about when in time they needed to accomplish each goal in relation to their temporal position in the game. If participants were solely remembering sequences, we would not expect to see behavioural and neural differences between the same goals as they changed relevancy over time. Further, naturalistic goals are often learned about more incidentally and more strongly integrated into one's autobiographical knowledge base. As discussed above, we chose our space-themed design to tightly control the experiential properties of the goals and investigate the impact of the timestamp attached to a goal alone. Additional studies are warranted to examine whether existing goals from participants' personal pasts and futures stratify to the same degree along the hippocampal long axis.

This research demonstrates that temporal distance plays a key role in guiding representations of personal goals, and this work has important implications for psychiatric disorders, namely depression. Depressed individuals generate more conflicting goals[55], envision more obstacles to achieving their goals and form less specific goals[56], feel less control over goal outcomes[57], and are more pessimistic regarding the likelihood of achieving their goals[58]. It is unclear whether there are inherent differences in how depressed individuals represent the temporal distance to different goals. It is possible that distance miss-representations contribute to differences in the perceived likelihood of success and specificity of future goals. If so, re-adjusting individuals' representations of time may offer therapeutic potential.

In conclusion, we demonstrate that the brain separates goals that are relevant for ones' current needs from those that are removed in time. Behaviourally, this is reflected in the reaction time differences to process current and temporally removed past and future goals.

Neurally, these sets of goals are dissociable along the posterior and anterior part of the hippocampal axis. We demonstrate that temporally removed goals are represented more anteriorly in the left hippocampus and current goals more posteriorly. Altogether, we provide evidence for an application of the hippocampus's long-axis system for attaching a timestamp to goals. These findings also provide insights into how time is represented in the human brain. Time is inherent to memory, decision making and hippocampal function, and evidence for a separate representation of "when" advances our mechanistic understanding of hippocampal function. Future research should further examine how different temporal representations are integrated in the hippocampus during memory and decision making.

## Methods

The Institutional Review Board at the Icahn School of Medicine at Mount Sinai approved this experiment. Compliance with all relevant ethical regulations was ensured throughout the study and methods were carried out in accordance with relevant guidelines and regulations.

### Participants

Thirty-four healthy participants completed the experiment from New York, NY, USA. Two participants were excluded for exaggerated head motion during the scan ( > 2.5 mm voxel size) and one participant was excluded for technical difficulties during acquisition. Thirty-one medically and psychiatrically healthy adults were included in the final analyses (Age; M = 27.0, SD = 4.5, Range=19–36, $n$ = 15 females). No statistical method was used to predetermine sample size. A sex analysis was not included given existing literature suggested that it would have no significant impact on the results of the study. All participants provided written informed consent and were financially compensated for their participation.

### Materials

Participants filled out 2 in-house developed screening questionnaires and a standard PDSQ (Psychiatric Diagnostic Screening Questionnaire[59]). The first questionnaire screened for pregnancy and any neurological, psychiatric, or substance use disorders. The second questionnaire screened for magnetic resonance imaging incompatibilities. Participants completed a post-task questionnaire after the scan.

### Procedures

The experiment took place over two days. Participants completed three screening questionnaires and a training paradigm on the first day ( ~ 1.5 h). They returned to the laboratory 24 hours later for their MRI scan. Participants were first familiarized with the scanning environment and experimental task in a mock scanner during this visit ( ~ 30 m). They then underwent their 7 T scan, where they completed the Mars task ( ~ 1 h), and filled out a post-task questionnaire after exiting the scanner ( ~ 15 m). See below for a comprehensive description of the training, mock scanner, and experimental paradigms.

**Training paradigm.** On the first day of the experiment, participants learned which goals they needed to accomplish in their 1st, 2nd, 3rd, and 4th years on Mars, which goals they always needed to accomplish, and which goals they never needed to accomplish. Goals fell under the categories of (1) maintaining a part of the space shuttle, (2) maintaining a part of their spacesuit, (3) eating a certain food, (4) completing a certain exercise, and (5) enjoying a recreational activity. There were 5 goals associated with each of the 6 timeframes (year 1, year 2, year 3, year 4, always, never) totalling to 30 stimuli. Participants learned the goals using a view and quiz procedure. During the viewing phase, the goals relevant for year 1 were shown for 3000 ms each, followed by a 1000 ms fixation cross. During the quizzing phase, participants were randomly shown the 5 goals for year 1, as well as several foil images (i.e., goals not belonging to year 1). If the goal presented belonged to year 1, participants had to press "1" and if the goal was a foil they had to press "0" on the keyboard. Participants had 6000 ms to respond and received feedback regarding whether they were correct for 1500 ms, followed by a 1000 ms fixation cross. If participants answered incorrectly or did not answer in time the quiz repeatedly restarted until they answered all questions correctly. Participants then advanced to viewing the 5 goals for year 2. During the year 2 quiz, participants were shown goals from year 1, year 2, and several foil images. They had to indicate "1" if the goal presented belonged to year 1, "2" if the goal presented belonged to year "2", or "0" if the image presented was a foil. This pattern continued such that participants were cumulatively quizzed on which years each of the goals belonged to.

Participants learned the goals in the following order: year 1, year 2, year 3, year 4, always, never in the first round of the training paradigm. They then completed the training in the opposite order to ensure that all goals were quizzed equally. The first 4 quizzes in each round contained between 2 and 4 foil images, the last 2 quizzes per round did not contain foils. Goals were randomly presented during both the viewing and quizzing phases. The training was programmed with E-prime 2.0 (https://pstnet.com) and stickers labeled "1", "2", "3", "4", "A", "N", and "0" covered keys S-K on the keyboard. Participants took approximately 45–90 min to complete this exercise. See Supplementary Table 10 for the instructions read and presented on the screen to the participants. The experimenter remained in the room to clarify any questions.

**Mock scanner training.** Participants were brought to a mock scanning room to be familiarized with the scanning environment and experimental task prior to their scan on the second day of the experiment. The experimenter explained to the participants that they would be embarking on a mission to Mars, and that once on Mars, the mission will last four years where they would be presented with the goals that they learned the day prior and will be asked when they need to accomplish each goal. Options included currently, in the near or distant future, in the near or distant past, always, or never. Near future was defined to the participants as 1 year in the future and distant future defined as 2 or 3 years in the future. Likewise, near past was defined to the participants as 1 year in the past and distant past defined as 2 or 3 years in the past. The experimenter emphasized that to succeed on the mission participants needed to think about each goal, when they had to complete it, and what Mars year they were currently in as they advanced through the mission. See Supplementary Table 10 for the full instructions delivered to the participants during this session.

Participants were then put in the mock scanner to complete an abbreviated version of the experimental paradigm. This version navigated participants through their four years on Mars, showing them 6 sample trials per year. The stimuli presented in this version were irrelevant (the foil images from the day prior). Participants were informed that they should not focus on the content of the trial, but rather use this exercise to familiarize themselves with the graphics of the task, transition slides, and button pressing hand pads.

**Experimental paradigm.** Participants then went on their mission to Mars in the 7 T scanner. Once participants are positioned in the scanner, a short movie is shown depicting the journey from Earth to Mars. Upon arrival, the Curiosity rover welcomes them (see Fig. 1b; QR code for the movie). The task then begins with a screen announcing that the participants are starting their first Mars year. Then, they were presented with a goal and asked when they need to complete it. Using the buttons on the bottoms of the screen they indicated if they needed to complete the goal in the current year (CY), near future (NF), distant future (DF), whether they always needed to complete it (AN), or whether they never needed to complete it (NN). Responses were followed by a 0.8 s or 2.5 s (randomly assigned) fixation cross before the onset

of the next trial. If participants were shown the space helmet, a goal that had to be completed in their 3rd year in version 1 of the task, they would select the DF button to indicate that they need to complete this goal in the distant future. Participants evaluated the temporal relevance of each of the 30 goals before seeing a screen that indicated that they had succeeded in completing their 1st year.

They then advanced to their 2nd year on Mars and a new button, NP, appeared to indicate goals that had been accomplished in the near past. Participants re-evaluated the temporal relevance of the 30 goals, i.e., they were shown the same 30 stimuli and had to reframe their thinking now that they had moved forward in time. Continuing with the example above, participants were shown the space helmet again, which still needed to be accomplished in their 3rd year. However, now that they were in their 2nd year on Mars, this goal must be completed in the near future, even though last year it was something that did not need to be completed until the distant future.

When participants transitioned to their 3rd year on Mars, another button, DP, appeared to indicate goals that had been completed in the distant past. The DF button was removed as there was only one year left in the mission. Participants re-evaluated the 30 goals such that the space helmet would now be a goal that must be accomplished in the current year, CY. When participants advanced to the 4th Mars year, the NF button was removed as they were in their final year and there was no longer a future component. Now, the space helmet became a goal that was accomplished the previous year, so participants would select NP to indicate that they completed this goal in the near past. See Supplementary Fig. 7 for an illustration of how the game buttons changed throughout the mission.

The key aspect of this design was that the same goals were presented each year, but their temporal distance changed as the participants advanced through the game. Goals that needed to be accomplished in the 1st year followed a temporal trajectory that moved from a current goal in the 1st year, to a near past goal in the 2nd year, a distant past goal in the 3rd year, and an even more distant past goal in the 4th year. Goals that needed to be accomplished in the 2nd year moved from near future goals, to current goals, near past goals, and distant past goals in the 4th Mars year. Goals that needed to be accomplished in the 3rd year transitioned from distant future goals, to near future goals, current goals, and near past goals in the 4th Mars year. And lastly, goals that needed to be completed in the 4th year moved from distant future goals to distant future goals, near future goals, and current goals in the final Mars year. Supplementary Table 11 presents a visual depiction of the temporal trajectories.

The mission contained 120 trials in total, which were divided into instances where participants were making an evaluation of a goal that needed to be completed in the distant future (15 trials), near future (15 trials), near past (15 trials), distant past (15 trials), current year (20 trials), always needed (20 trials), and never needed (20 trials). These trials were distributed across the four Mars years and a visual illustration of this distribution is provided in Supplementary Table 2. Participants used their index through ring fingers to press the buttons on two hand pads. The buttons rotated on the screen after every trial to prevent the reaction times for a condition from being influenced by the ease of using one finger over another. The task was self-paced, and participants did not receive feedback on their selections. Between each year, participants were given the opportunity to pause and press a button when they were ready to advance. Participants took on average 17 min to complete the task (M = 17.79 min, SEM = 0.72, Mdn = 17.45 min, Range = 13.08–33.76 min). The task was programmed in Unity version 5.6.3f1 (https://unity.com/).

**Stimuli selection.** Goal stimuli were selected using an independent online survey. Individuals were asked to rate the valence, arousal, and familiarity of the goals using a sliding scale from 0 [low] – 9 [high] ($n = 16$ for the space shuttle, food, exercise, recreational activity goals, $n = 9$ for

the spacesuit goals). We visually examined that the goal ratings were neutral, i.e., no goals were viewed as extremely high or low in valence, arousal, or familiarity. Goals from each category were randomly assigned to sets, such that each set of stimuli had a space shuttle, space suit, food, exercise, and recreational activity goal. Each set of goals stayed as a unit for this point forward, but the timeframe that the goal set was assigned to (year 1, year 2, year, 3, year 4, always, never) varied by task version. There was no statistically significant difference in valence, arousal, and familiarity across each set of goals (Supplementary Fig. 8, Kruskal-Wallis rank sum test; valence: $X^2_{(5)} = 3.02$, $p = 0.70$, arousal: $X^2_{(5)} = 6.11$, $p = 0.30$, familiarity: $X^2_{(5)} = 2.05$, $P = 0.84$). The surveys were administered via SurveyMonkey (https://www.surveymonkey.com).

**Task versions.** We created 6 versions of the task such that each set of goals was associated with every timeframe (to be accomplished in year 1, year 2, year, 3, year 4, always, never). Each set of 5 goals stayed as a unit. However, a set needed to be accomplished in year 1 for participants completing version 1 of the task, in year 2 for participants completing version 2 of the task and never for participants completing version 6 of the task. See Supplementary Fig. 9 for an illustration of the stimuli and version system and Supplementary Table 12 for a distribution of participants by version.

**Additional choice trials.** Randomly throughout the mission, participants were presented with two goals on the screen and instructed to choose the goal they felt would most benefit their survival at the present moment. The choice was between a goal relevant for the current year and a goal relevant for another timeframe, and participants indicated their choice by clicking either the "Left" or "Right" thumb buttons. They were presented with 100 choices in total (25 per year), with a current item being compared to a distant (30 trials) future (15 trials), near future (15 trials), near past (15 trials), distant past (15 trials), always needed (20 trials), and never needed (20 trials) goal (see Supplementary Fig. 3 for results).

**Post-task questionnaire.** Participants filled out a questionnaire immediately after exiting the scanner. This questionnaire asked participants whether they were hungry during the scan, motivated to complete the task, whether motivation lessened at any point during the task, whether they employed a strategy to remember the goals, whether they felt time was passing during the mission, and lastly, a free recall task (to write down as many goals as they could remember). An analysis of this data is not included.

### Neuroimaging methods

**fMRI acquisition.** Data was acquired on a 7 T Siemens Magnetom scanner (Erlangen, Germany) with a 32-channel head coil (Nova Medical, Wilmington, MA). Anatomical images were collected with a twice magnetization-prepared rapid gradient echo (MP2RAGE) sequence for improved T1-weighted contrast and spatial resolution (Marques et al., 2010) [0.7 mm isotropic resolution, 224 slices, TR = 6000 ms, TE = 5.14 ms, field of view=320×320, bandwidth=130 Hz/Px]. Functional data was acquired in a single run using a multi-echo multi-band echo-planar imaging (EPI) pulse sequence [2.5 mm isotropic resolution, 50 slices, TR=1850ms, Tes= [8.5, 23.17, 37.84, 52.51 ms], MB = 2, iPAT acceleration factor = 3, flip = 70, field of view=640 × 640, pixel bandwidth=1786]. This acquisition approach situated us in a prime position to detect small biological effects. Ultra-high field 7 T fMRI provides 2x the signal-to-noise ratio as traditional 3 T fMRI (Kundu et al., 2017), and multi-echo protocols maximize BOLD contrast throughout the brain and offer 5-fold increases in statistical power over singe-echo acquisitions (Kundu et al., 2012, 2017).

**fMRI data preprocessing.** Functional files were denoised for physiological and motion artifacts using multi-echo independent

components analysis (ME-ICA, https://bitbucket.org/prantikk/me-ica). A detailed description of this pipeline can be found here (Kundu et al., 2012, 2017). In short, ME-ICA is based on the observation that BOLD signals have linearly TE-dependent percent signal changes, as a result of T2* decay, while artifacts do not. ME-ICA decomposes multi-echo functional data into independent components and computes the TE dependence of the BOLD signal for each component. It categorizes components as either BOLD or non-BOLD and removes the non-BOLD components, leaving the data robustly denoised of motion, physiological, and scanner artifacts. As shown in Morris, et al[60]., the use of the ME-ICA pipeline further enhances the signal power gains obtained when moving from 3 T to ultra-high field 7 T fMRI.

The ME-ICA pipeline outputted a denoised timeseries with T1 equilibration correction (including thermal noise) in the participant's native space and the remaining preprocessing steps were performed using the Functional MRI of the Brain Software Library (FSL) version 5.0.10[61]. Anatomical images were first skull-stripped using robust brain centre estimation in BET. Functional images were registered to the participant-specific high resolution T1-weighted structural images using boundary-based registration[62] and to a standard brain image (MNI 152 T1 2.5 mm³) using a 12 DOF linear registration in FSL's fMRI Expert Analysis Tool, FEAT[63]. Since the task was self-paced and the scans manually stopped, registered functional files were trimmed to include only 7 volumes following each participant's completion of the task. Images were spatially smoothed at 5mm FWHM (double the voxel size) to ensure GRF theory validity and this is the most appropriate smoothing for smaller structures such as the hippocampus[64,65].

**Regions of interest.** We defined individual right and left hippocampal masks using FreeSurfer v7.2, we segmented the hippocampal subfields and nuclei of the amygdala (Iglesias et al., 2015). We first used the 'recon-all' and 'autorecon1' functions to get the t1 whole brain scan for each participant. Then we used the hippocampus segmentation tool ('segmentHA_t1.sh'), we reoriented the masks using 'fslreorient2std', we merged the different parts of the hippocampus (i.e., CA1, CA2, etc.) using 'fslmaths'. Masks were registered applying the transformation estimated between the T1 and the 2.5 mm³ brain template using FSL ('flirt' and 'applyxfm'). Finally, the group average ROI for hippocampus subportions was obtained by binarizing and averaging individually drawn ROIs for all participants, and selecting only voxels common to at least 80% of the participants. We created a whole brain gray matter mask by resampling SPM12's gray matter tissue probability map to the 2.5 mm³ space and thresholding with a probability threshold of 20%. Supplementary Fig. 10 visualizes all masks.

**Analyses**
**Behavioural analyses.** Participants' reaction time data was analysed using a series of linear mixed models, implemented using the lme4 and lmerTest packages in R v. 3.6.0. The data was first processed to remove trials where the participant got the question incorrect, and trials considered outliers (reaction times +/− 3 SDs of a participant's mean). This left 94.5% of all trials across the 31 participants. Reaction times were log transformed prior to modelling. The first model examined the relationship between participants' log transformed reaction times as the dependent variable and temporal condition as the independent variable. Temporal condition was modelled as a fixed effect with five levels: distant future, near future, near past, distant past, and current trials. We removed always and never trials from the data set for this model because they inherently differed in that they consisted of the same stimuli each game year, while the stimuli in the other categories updated as time progressed.

We validated that the results held in a second model that included the always and never trials. The second model examined the

relationship between participants' log transformed reaction times as the dependent variable and trial type as the independent variable. Trial type was modelled as a fixed effect with two levels. Distant future, near future, near past, and distant past trials were coded as those involving a temporal component, while current, always, and never trials were coded as those not.

Participant ID was included as a random effect to account for the lack of independence between trials within a participant, and game year was included as a fixed effect and as a random effect for objective time (game year) to account for trends in reaction time across the years as the mission proceeded within and between participants. This game year regressor had 4 levels (year 1, year 2, year 3, year 4) and was in place to absorb variance such that any reaction time differences that remained between the temporal conditions were those above and beyond what could be explained by objective time point in the game.

Statistical significance was evaluated using Type II Sums of Squares. Model-based estimated marginal means, standard errors, confidence limits, and post hoc tests were computed via the emmeans package in R. Post hoc tests compared two estimated marginal means and outputted the corresponding estimate, standard error, test statistic, and p value (for a detailed list of the R packages utilized in our analyses, including their versions and citation information, refer to the Supplementary Information Materials section). Degrees of freedom were approximated using the Satterthwaite method and post hoc tests were corrected for multiple comparisons using a Tukey adjustment. See Table 1 for the model formulas.

**Neuroimaging analyses**
**General linear modeling analyses.** Univariate analyses consisted of general linear models implemented in FSL's FEAT[61]. General temporal analysis: For the first-level analyses, models included four separate regressors, one for each temporal condition, including only correct responses: remote, current, always, and never. Temporal gradient analysis: For the first-level analyses, models included seven separate regressors, one for each temporal condition, including only correct responses: distant future, near future, near past, distant past, current, always, and never. General temporal and temporal gradient analyses: Regressors for each game year (year 1, year 2, year 3, year 4) were also included in the model. These contained all trials in the year specified, i.e., the task, additional choice, incorrect responses, and reaction time outlier trials were in place to absorb variance explained by position in the task. The duration of each trial was set as the participant's reaction time on that trial due to the self-paced nature of the experiment. All trials were modeled with boxcars and convolved with FEAT's canonical hemodynamic response function. This approach captures the temporal dynamics of neuronal processes underlying the task, as supported by previous research[66] and as mentioned in a widely used reference handbook in the fMRI community[64]. Temporal derivatives of the aforementioned regressors and six motion regressors were also included in the model. See Supplementary Table 4 (general temporal analysis) and Supplementary Table 5 (temporal gradient analysis) for all regressors included in the analysis. A high pass filter with a cutoff of 128 s was applied to remove low frequency signal drifts.

**General temporal analysis.** Two contrasts compared the temporally removed condition to the current condition (remote > current, and current > remote). Temporal gradient analysis: eight contrasts compared the temporally removed conditions to the current condition (distant future > current, near future > current, near past > current, distant past > current, and the reverse). First level activation maps were brought to a second level mixed effects analysis, implemented in FLAME 1 (FSL's Local Analysis of Mixed Effects), where one sample t-tests were used to determine the group mean for each

contrast. Statistical maps were masked as described above and results corrected for multiple comparisons using Gaussian random field theory[67].

**General Psychophysiological Interaction (gPPI) analyses.** We used FSL's FEAT to investigate connectivity effects between the anterior/posterior seed regions and other brain regions under the remote and current conditions (**General temporal analysis**). The gPPI method, as proposed by[68] was employed to extend the classic PPI analysis. This method generates a PPI term for each condition in the study, along with the seed time-series regressor and individual regressors for each condition.

We used the fMRI data after preprocessing and the hippocampal regions of interest (ROIs) that were already defined for the GLM analysis. The same event-related time points used in the GLM were used for the gPPI. The average of the preprocessed functional data within the hippocampal ROIs was taken as the seed region time series for each participant. For each condition, a PPI term was created. First level activation maps were brought to a second level mixed effects analysis, implemented in FLAME 1 (FSL's Local Analysis of Mixed Effects), where one sample t-tests were used to determine the group mean for each contrast.

Hippocampal, whole brain GLMs, and gPPI were corrected for multiple comparisons using GRF-theory-based maximum height thresholding with a family wise error (FWE) corrected significance threshold of $p = 0.025$. Activation maps were visualized using MRIcroGL's suite of visualization tools[69] (https://www.nitrc.org/projects/mricrogl).

### Reporting summary

Further information on research design is available in the Nature Portfolio Reporting Summary linked to this article.

## Data availability

The fMRI and behavioral data generated in this study have been deposited in the Open Science Framework repository database[70] [https://doi.org/10.17605/OSF.IO/3KT98].

## Code availability

The behavioral code generated for this study has been deposited in the Open Science Framework repository[70] [https://doi.org/10.17605/OSF.IO/3KT98].

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

## Acknowledgements

The authors thank Dominik Moser for his help with the pilot studies, Matthew Schafer and Ofer Perl for their helpful discussions, Michaël Dayan, Ben Meulman for their help with MRI and behaviour statistics, and Yael Jacobs for her assistance implementing the fMRI preprocessing pipeline. We would also like to thank Thomas Grand from Atelier XL for illustrating the stimuli and creating the movie, and Adrienne Young for assisting with the Unity programming. D.S. is supported by the National

Institute of Health, USA (R01MH122611, R01MH123069); A.M. is supported by a Swiss National Science Foundation Grant (P2GEP1-165097); D.E.C. is supported by an NSF graduate research fellowship; MGP is supported by the CIBM Center for Biomedical Imaging, a Swiss research center of excellence founded and sup- ported by Lausanne University Hospital (CHUV), University of Lausanne (UNIL), Ecole polytechnique fédérale de Lausanne (EPFL), University of Geneva (UNIGE) and Geneva University Hospitals (HUG).

## Author contributions

A.M. developed the research question; A.M., D.E.C., and D.S. designed the research; L.L programmed the task; D.E.C. collected the data; A.M., and D.E.C. analysed the data with the input from M.G.P, M.G.B, and D.S. A.M. and D.E.C. wrote the manuscript with input from all authors.

## Competing interests

The authors declare no competing interests.
