## [Peer Review File · Nature Communications]

The hippocampus dissociates present from past and future goalsEditorial Note: This manuscript has been previously reviewed at another journal that is not operating a transparent peer review scheme. This document only contains reviewer comments and rebuttal letters for versions considered at *Nature Communications*. Mentions of the other journal have been redacted.

Reviewers' comments:

Reviewer #1 (Remarks to the Author):

I reviewed a previous version of this manuscript when it was submitted to a different [redacted] journal. I appreciate the thoughtfulness of the authors' responses to my prior comments, which were addressed completely. Their updated analyses make a compelling case for the stated conclusions, and I think the manuscript will be of interest to a wide audience. I would suggest only two very minor clarifications based on the revised text.

1) Lines 244-245 state a $y = -21$ boundary for Remote > Current goal activations within the hippocampus. The authors may wish to state here, rather than in lines 258-260, that this coordinate is significant because it marks a standard boundary of anterior vs. posterior hippocampus.

2) Would it be possible to include a difference map in Supplementary Figure 5 to better illustrate the anterior/posterior connectivity differences summarized by the authors in response to my prior point #6 and in their new figure text?

Reviewer #3 (Remarks to the Author):

The authors are to be commended for doing a good job addressing the issues that I raised as part of the review process at a previous journal. In particular, they have toned down their claims with regards to the granularity of current vs. distant goal representations and solidified a number of aspects with regards to their neuroimaging analyses.

While the paper is much approved, I have a few remaining suggestions that may potentially provide deeper insight into what is driving the anterior vs. posterior hippocampal activity differences that have been observed, and thereby, strengthen the interpretation of the reported findings.

1) I am curious to know whether a gPPI analysis would yield any significant findings, using the most significant anterior/posterior voxels as seeds? This would add significantly more than the reported preliminary functional connectivity analysis that has been reported in the supplementary materials.

2) Was there a significant RT difference between 'current' trials and 'never' trials, as well as between 'current' trials and 'always' trials? There was a clear difference between trials with temporal computation and those without (e.g., Supplementary Figure 1) but it is unclear to what extent this was driven by the 'distant past/near past/near future/distant future' trials. The results of the aforementioned comparisons may have implications for the authors' discussion from line 363 to 370.

3) Assuming I have interpreted Supplementary Table 5 correctly (line 605 states that this table refers to results from the additional choice trials but the table appears to describe neuroimaging contrast findings?) I find it intriguing that the AN > Current contrast revealed greater anterior and posterior hippocampal activity, while the NN > Current contrast revealed greater anterior hippocampal activity. The latter is not inconsistent with the authors' suggestion regarding abstract representation and anterior hippocampal involvement, and indeed the greater anterior hippocampal activity for the contrast Remote > NN also fits with this. How does one account for the findings of the former

contrast, when one would expect both current and always needed goals to be represented in detail?

4) The authors suggest that the behavioural pattern on the discrimination trials (line 206 - 216) may reflect pattern separation (and related to the above, it's unclear where these behavioural data are) - do the imaging data provide any insight into this? Currently, no imaging analyses are reported with respect to these trials and it is unclear why not (apologies if I have just missed something).

Reviewer #4 (Remarks to the Author):

In this paper, entitled "Hippocampal timestamp for goals" by Montagnin et al, the authors present a behavioral and functional imaging study, while participants are asked to make decisions about goals in relation to their timing. They find that participants are faster at making goal temporal decisions (when is this happening?) for current goals compared to those that are not current. In the fMRI data, they present a contrast between current goals and those that are not current and report one anterior hippocampal region that is more active for 'not current' decisions compared to current goal decisions. By contrast, they show a distinct cluster of voxels that they report to be in 'posterior hippocampus' (although it looks to be in the body and tail, see below) that is more active for current compared to not current goal decisions. The manuscript is focused on a framework that introduces levels of granularity across the long axis of the hippocampus, and the interpretation of the paper seems to suggest that time may also be represented in a similarly granular fashion. However, the results in my view do not present strong evidence for this interpretation.

I find the design really clever and well suited to address the question of whether temporal proximity or time is represented along the axis of the hippocampus. However, the results seem to indicate a distinction between two rather small, hippocampal peak voxels that show statistically opposite effects, the left anterior hippocampal region shows greater activation for all non-current goals compared to current and a different region (in the body and tail of the hippocampus) shows greater activation for current decisions compared to non-current. From Table 2, there is also another region in more mid hippocampus that shows a remote > current, but on the right side.

My main concern is what can we take away from these results. A conservative interpretation would be that task demands between these two different trials led to different statistical effects in small hippocampal (and cortical) clusters. Is there anything else that distinguishes these two trials besides their current vs remote goal factor? Yes, the authors present really clear evidence that response times differ significantly between 'current' and all other goal decisions. It is very possible that response times contribute to the activations seen and any fMRI analyses should take time-on-task into account. This is highly recommended as a supplemental analysis to use response times on every trial as a regressor in the GLM.

The interpretation chosen by the authors is that the hippocampus somehow represents current goal and distant goals in a manner consistent with emerging evidence in the literature for a difference in 'granularity' across the hippocampal axis. But the analyses and approach offered in this revised manuscript is not looking for a gradual change with temporal distance which is what the Introduction sets us up to expect.

I was surprised not to see any representational similarity analyses conducted in this paper, making for a rather 'thin' results section. Identifying two clusters based on GLMs showing univariate differences across conditions with no follow up to gauge what kind of information or representations are supported in those regions is surprising for an imaging paper in 2023.

On a scholarly note, I found the Intro to be lacking in mechanistic depth. There are many statements that refer to papers and how they show evidence for different effects but the descriptions do not detail what that evidence is, one is left wondering if the authors read the papers cited. I recommend offering

an informed synthesis of the cited work, where possible.

In sum, I want to commend the authors on a clever paradigm but the results and analyses are lacking any strong evidence for a 'timestamp'. I do not anticipate these results will have an impact on our understanding of time, goals or hippocampal function.

Dear editor and reviewers,

Thank you for the additional comments on the revision. We believe we were able to address the concerns in full, as detailed below. We indicate the comments followed by our responses and the corresponding changes in the text.

Reviewer #1:

I reviewed a previous version of this manuscript when it was submitted to a different [redacted] journal. I appreciate the thoughtfulness of the authors responses to my prior comments, which were addressed completely. Their updated analyses make a compelling case for the stated conclusions, and I think the manuscript will be of interest to a wide audience. I would suggest only two very minor clarifications based on the revised text.

1) Lines 244-245 state a $y = -21$ boundary for Remote > Current goal activations within the hippocampus. The authors may wish to state here, rather than in lines 258-260, that this coordinate is significant because it marks a standard boundary of anterior vs. posterior hippocampus.

Action taken: Following to the reviewer's suggestion, we now state: "The contrast remote > current identified voxels falling anterior to $Y=-21$ (mm in MNI coordinate). Current goals produced stronger activation in the left posterior hippocampus, with voxels extending from $Y=-31$ to $Y=-36$ (**Figure 3b; Table 2**). These voxel locations align with the posterior and anterior boundaries outlined in Poppenk et al. (2013), who defined the foci of the anterior hippocampus to $y = -21$, using the uncus apex as the anatomic landmark." (pg. 14, lines 248-253).

2) Would it be possible to include a difference map in Supplementary Figure 5 to better illustrate the anterior/posterior connectivity differences summarized by the authors in response to my prior point #6 and in their new figure text?

Response: In response to a comment raised by another reviewer, we conducted gPPI analyses to address the question of connectivity. We have decided not to continue with CAPs analyses because they were exploratory in nature and not specific to conditions. Given the specific focus on connectivity per conditions, we believe that the gPPI analyses provide a more relevant and targeted approach for addressing this aspect of our study.

Action taken: We now include maps of functional connectivity for temporally removed and current goals with the anterior or posterior hippocampus as seeds (Supplementary Figure 6, Tables 7 and 8).

Reviewer #3:

The authors are to be commended for doing a good job addressing the issues that I raised as part of the review process at a previous journal. In particular, they have toned down their claims with regards to the granularity of current vs. distant goal representations and solidified a number of aspects with regards to their neuroimaging analyses.

While the paper is much approved, I have a few remaining suggestions that may potentially provide deeper insight into what is driving the anterior vs. posterior hippocampal activity

differences that have been observed, and thereby, strengthen the interpretation of the reported findings.

1) I am curious to know whether a gPPI analysis would yield any significant findings, using the most significant anterior/posterior voxels as seeds? This would add significantly more than the reported preliminary functional connectivity analysis that has been reported in the supplementary materials.

Response: Following the advice of the reviewer, we used the most significant voxels as seeds. However, this analysis did not yield any significant connectivity for the anterior peak voxel. Utilizing a single voxel possibly introduced more noise and variance, thereby diminishing the accuracy and strength of the observed interaction effects. We therefore performed gPPI with the anatomical masks (the same ones used for the ROI analyses). We found that when goals were in the current moment, the left posterior hippocampus showed stronger association with regions in the right hemisphere, while the anterior part was primarily linked to regions in the left hemisphere. However, when goals were removed in time, both the left posterior and anterior hippocampus exhibit a greater association with regions in the right hemisphere. This pattern suggests a dynamic shift in hemisphere involvement based on the temporal context of the goals.

Action taken: We now report the connectivity maps in Supplementary Figure 6, and Table 7 and 8 (see below).

Supplementary Figure 6. Analysis of functional connectivity for temporally removed and current goals. We performed a General Psychophysiological Interaction (gPPI) analyses, as seed regions, we used left anterior and posterior hippocampus. For the goals that were current (a) the posterior part of the hippocampus activated the right hemisphere, whereas (b) the anterior part activated the left hemisphere. For the goals that were removed in time (c) The posterior part of the hippocampus and (d) the anterior part of the hippocampus activated the right hemisphere. All z-statistic images were thresholded parametrically using maximum height thresholding (FWE voxel-wise correction, $p=0.025$). Contrasts maps were overlaid and rendered onto a 3-dimensional MNI 152 brain using MRICroGL. Color bars reflect the thresholded z-statistic scores [$z=0-4$]. R, right; L, left.

a

Voxel size	Hemisphere	Region	Z-stat	Coordinates in mm
40	R	Frontal lobe	5.9	30 14 60.5
38	R	Parietal lobe	6.64	50 -43.5 50.5
12	L/R	Frontal lobe	5.41	0 26.5 43
9	R	Frontal lobe	5.39	43 42 22

b

Voxel size	Hemisphere	Region	Z-stat	Coordinates in mm
3	R	Frontal lobe	5,32	37.5 39 23
3	R	Cingulate gyrus	5	7.5 24 40.5
3	R	Parietal lobe	5,18	42.5 -31 65.5
2	R	Parietal lobe	5,19	45 -41 48
1	R	Cerebellum	5,01	0 -56 -34.5
1	R	Parietal lobe	5,05	-43.5 50.5 25.6

Supplementary Table 7. General Psychophysiological Interaction (gPPI) Analysis Results and Coordinates for Posterior Hippocampus Connectivity. This table reports the voxel coordinates, the maximum z-statistic within the cluster, and the x, y, z location of the significant functional connectivity observed in the posterior mask of the hippocampus for the (a) current condition, and (b) remote condition. Coordinates are reported in Montreal Neurological Institute (MNI) space. All statistical maps were corrected for multiple comparisons using maximum height thresholding (FWE voxel-wise correction, $p=0.025$).

a

Voxel size	Hemisphere	Region	Z-stat	Coordinates in mm
10	L	Frontal Lobe	5.39	-27.5 -1 63
7	L	Frontal lobe	5.45	-20 -16 53
6	L	Frontal lobe/Parietal lobe	6.11	-37.5 -26 38
6	L	Frontal lobe	5.3	-42.5 -6 25.5
4	R	Frontal lobe	5.4	22.5 -16 50.5
5	L	Parietal lobe	5,33	-20 -53.5 40.5
3	L	Frontal lobe	5,31	-25 -11 50.5
1	L	Occipital Lobe	4,99	-10 -71 -7
1	L	Occipital Lobe	4,91	-10 -76 -2
1	L	Frontal lobe	5,14	-55 4 30.5

b

Voxel size	Hemisphere	Region	Z-stat	Coordinates in mm
1	R	Frontal lobe	4.88	52.5 9 30.5

Supplementary Table 8. General Psychophysiological Interaction (gPPI) Analysis Results and Coordinates for Anterior Hippocampus Connectivity. This table reports the voxel coordinates, the maximum z-statistic within the cluster, and the x, y, z location of the significant functional connectivity observed in the anterior mask of the hippocampus for the (a) current condition, and (b) remote condition. Coordinates are reported in Montreal Neurological Institute (MNI) space. All statistical maps were corrected for multiple comparisons using maximum height thresholding (FWE voxel-wise correction, $p=0.025$).

2) Was there a significant RT difference between 'current' trials and 'never' trials, as well as between 'current' trials and 'always' trials? There was a clear difference between trials with

temporal computation and those without (e.g., Supplementary Figure 1) but it is unclear to what extent this was driven by the 'distant past/near past/near future/distant future' trials. The results of the aforementioned comparisons may have implications for the authors' discussion from line 363 to 370.

Response: To address the reviewer's question, we now report RT differences among the trials without temporal computation. We found that 'never' trials were processed faster than 'current' and 'always', while there was no significant difference between 'current' and 'always'. These results indicate that participants were quicker to process goals that are irrelevant (never needed) versus those that are relevant (currently or always).

Action taken: The below figure (following the next comment) has been added to the supplementary material as Supplementary Figure 2, and has been referenced in the main text (page 15, line 262). In addition, we have updated the discussion as follows: "Regarding conditions without temporal constraints, goals categorized as never relevant exhibited swifter processing compared to goals designated as current or always relevant. This observation highlights a possible mechanism that governs the adjustment of goal processing based on temporal context (when goals matter) and relevancy (do goals matter)." (pg. 20, line 384-388).

3) Assuming I have interpreted Supplementary Table 5 correctly (line 605 states that this table refers to results from the additional choice trials but the table appears to describe neuroimaging contrast findings?) I find it intriguing that the AN > Current contrast revealed greater anterior and posterior hippocampal activity, while the NN > Current contrast revealed greater anterior hippocampal activity. The latter is not inconsistent with the authors' suggestion regarding abstract representation and anterior hippocampal involvement, and indeed the greater anterior hippocampal activity for the contrast Remote > NN also fits with this. How does one account for the findings of the former contrast, when one would expect both current and always needed goals to be represented in detail?

Response: Thank you for pointing out that the table was incorrectly referenced in the manuscript. This is now corrected (we have placed the table within Supplementary Figure 2). Regarding the interpretation of the neural results for trials without temporal computation, it is important to note that we do not make claims about levels of details or abstractness, as these components were deliberately removed from the stimuli. We can only refer to the representation of the temporal component of the goals. In the case of Always vs Current, these trials did not differ significantly in reaction time, but the fact that Always trials had stronger representation in both anterior and posterior parts, perhaps stems from the fact that, unlike Current trials, Always trials are not only relevant to the current time, but also to the past and to the future, and thus may have a broader representation.

Action taken: We added the interpretation of these results to the legend of Supplementary Figure 2.

Supplementary Figure 2. a) Post hoc reaction time comparisons for current versus always needed (AN) and never needed (NN) and b) coordinates of peak activation for temporally removed, current, always needed and never needed goals in the hippocampus. a) Post hoc comparisons of the estimated marginal means revealed that current and AN goals were processed slower than NN goals. Error bars = 95% CI. C, current goals; A, always needed goals; N, never needed goals. **b)** Always needed (AN) goals activated both the posterior and anterior regions. This suggests that, unlike Current trials, Always trials are not only relevant to the current moment, but also to the past and to the future, and thus may have a broader representation. The never needed (NN) goals exhibited a resemblance to the remote goals, revealing activity in the anterior part of the hippocampus. The table reports the number of significant voxels in the cluster, maximum z-statistic within the cluster, and the x, y, z location of the maximum intensity voxel for each contrast. Coordinates are reported in Montreal Neurological Institute (MNI) space. All statistical maps were corrected for multiple comparisons using maximum height thresholding (FWE voxel-wise correction, $p=0.025$). AN, always needed goals; NN, never needed goals.

4) The authors suggest that the behavioural pattern on the discrimination trials (line 206 - 216) may reflect pattern separation (and related to the above, it's unclear where these behavioural data are) - do the imaging data provide any insight into this? Currently, no imaging analyses are reported with respect to these trials and it is unclear why not (apologies if I have just missed something).

Response: The behavioural data are presented in Supplementary Figure 3. These trials were not intended for neuroimaging analysis. They are free choice, self-paced trials with reaction times around 0.5-1.5 second, and were used to test the behavioral hypothesis only. The results were consistent with our hypothesis and support the main conclusion, that goals that are closer in time are more difficult to separate as opposed to goals that are further apart in time. The pattern separation idea is speculative, and we suggest this idea for future studies.

Reviewer #4:

In this paper, entitled “Hippocampal timestamp for goals” by Montagrín et al, the authors present a behavioral and functional imaging study, while participants are asked to make decisions about goals in relation to their timing. They find that participants are faster at making goal temporal decisions (when is this happening?) for current goals compared to those that are not current. In the fMRI data, they present a contrast between current goals and those that are not current and report one anterior hippocampal region that is more active for “not current” decisions compared to current goal decisions. By contrast, they show a distinct cluster of voxels that they report to be in “posterior hippocampus” (although it looks to be in the body and tail, see below) that is more active for current compared to not current goal decisions. The manuscript is focused on a framework that introduces levels of granularity across the long axis of the hippocampus, and the interpretation of the paper seems to suggest that time may also be represented in a similarly granular fashion. However, the results in my view do not present strong evidence for this interpretation.

I find the design really clever and well suited to address the question of whether temporal proximity or time is represented along the axis of the hippocampus. However, the results seem to indicate a distinction between two rather small, hippocampal peak voxels that show statistically opposite effects, the left anterior hippocampal region shows greater activation for all non-current goals compared to current and a different region (in the body and tail of the hippocampus) shows greater activation for current decisions compared to non-current. From Table 2, there is also another region in more mid hippocampus that shows a remote > current, but on the right side.

My main concern is what can we take away from these results. A conservative interpretation would be that task demands between these two different trials led to different statistical effects in small hippocampal (and cortical) clusters. Is there anything else that distinguishes these two trials besides their current vs remote goal factor? Yes, the authors present really clear evidence that response times differ significantly between “current” and all other goal decisions. It is very possible that response times contribute to the activations seen and any fMRI analyses should take time-on-task into account. This is highly recommended as a supplemental analysis to use response times on every trial as a regressor in the GLM.

Response: We would like to clarify that the small number of the observed hippocampal peak voxels does not diminish the significance of our results. We employed a 7T scanner, which offers ultra-high resolution and improves signal-to-noise ratio, allowing for more precise localization of brain activations. Furthermore, we rigorously corrected for multiple comparisons ensuring that the reported effects are unlikely due to chance.

Regarding the role of reaction time (RT), we agree with the reviewer that it is important to account for RT as an alternative explanation, and we have done exactly so. In our analysis, we utilized a variable epoch model, convolving each event with a boxcar function equal to the length of the subject's response time for that trial. This approach captures the temporal dynamics of neuronal processes underlying the task, as supported by previous research (Grinband et al., 2008) and as mentioned in a widely used reference handbook in the fMRI community (Poldrack et al., 2011). We added additional explanation for this method in the manuscript (page 36, line 743-746).

Furthermore, not all conditions with shorter RTs appear anteriorly. For example, while RT to never needed goals is shorter than for current goals, the neural representation of never needed

goals is anterior to current goals rather than more posterior as one would have expected if RT were a confounding factor (Supplementary Figure 2). This observation suggests that RT alone cannot fully reflect the underlying brain activations.

The interpretation chosen by the authors is that the hippocampus somehow represents current goal and distant goals in a manner consistent with emerging evidence in the literature for a difference in “granularity” across the hippocampal axis. But the analyses and approach offered in this revised manuscript is not looking for a gradual change with temporal distance which is what the Introduction sets us up to expect.

Response: The reviewer is correct; we did not report a gradual change but rather a dissociation between current and temporally removed goals. Our revised introduction and edits to the discussion now more accurately reflect that, and we have removed any mentioning of “granularity.” We did examine the gradual change with temporal distance among the conditions of near, distant past, future, and current, but this analysis didn’t yield any new insights and was consistent with the main results. At the recommendation of the other reviewers, these findings are reported in Supplemental Table 5.

I was surprised not to see any representational similarity analyses conducted in this paper, making for a rather “thin” results section. Identifying two clusters based on GLMs showing univariate differences across conditions with no follow up to gauge what kind of information or representations are supported in those regions is surprising for an imaging paper in 2023.

Response: We appreciate the reviewer's suggestion regarding the potential use of representational similarity analyses (RSA) in our study. We would like to provide context and clarity on our methodological decisions regarding the analyses presented in this manuscript. In our initial submission, the study design encompassed multiple temporal conditions: distant past, close past, close future, and distant future. However, based on feedback from reviewers in the previous round, we were advised to consolidate these conditions into a single “temporally removed” category. This revised category was subsequently contrasted against the “current” condition using a GLM for the following reasons: (1) Enhanced statistical power: The merged “temporally removed” category allowed for increased statistical robustness due to a larger number of trials. (2) Mitigation of temporal confounds: Combining the conditions helped reduce specific biases that might be tied to any particular temporal point, ensuring a more representative depiction of non-present thinking. (3) Clear contrast: The direct GLM contrast between “temporally removed” and “current” furnishes an unambiguous, interpretable distinction between present moment thinking and any other time.

Regarding the suggestion to use of Representational Similarity Analysis (RSA): While RSA undoubtedly offers valuable insights in certain scenarios, our consolidated design, as per the earlier reviewers' recommendations, naturally aligns with the strengths of a univariate approach. With our focus on broad differences between the two main conditions, the GLM was adept at capturing these contrasts effectively. In this context, RSA's added nuances might not substantially enhance our primary research question, which explores the overarching differences between thinking of the present and other times. We will be happy to add such analysis if the editor and reviewer deem this appropriate.

In summary, our methodological decisions, shaped by earlier feedback and in line with our research objectives, aimed to offer clarity and robustness in our findings. We hope this clarifies the evolution of our analytical approach and the rationale behind it.

On a scholarly note, I found the Intro to be lacking in mechanistic depth. There are many statements that refer to papers and how they show evidence for different effects but the descriptions do not detail what that evidence is, one is left wondering if the authors read the papers cited. I recommend offering an informed synthesis of the cited work, where possible.

Response: We have rewritten large parts of the introduction to better convey the premise of the study and the hypotheses.

In sum, I want to commend the authors on a clever paradigm but the results and analyses are lacking any strong evidence for a “timestamp”. I do not anticipate these results will have an impact on our understanding of time, goals or hippocampal function.

Response: We appreciate the reviewer’s acknowledgment of our clever paradigm. However, we respectfully disagree with the reviewer’s opinion regarding the impact of the study. While previous research confounded time with degree of details and abstractness, this is the first study to clearly isolate the time component. We show that when holding all other factors fixed, time alone has a representation along the long axis of the hippocampus, with current goals represented more posteriorly and temporally removed goals represented more anteriorly. We have demonstrated this using state-of-the-art ultrahigh 7T neuroimaging, rigorously analyzed, corrected for multiple comparisons, and accounted for reaction time differences. Time is inherent to memory, decision making, and hippocampal function, and thus evidence for a separate representation of “when” advances our mechanistic understanding of hippocampal function, promoting future research to further examine how different representations are integrated in the hippocampus during memory and decision making.

References:

Grinband J, Wager TD, Lindquist M, Ferrera VP, Hirsch J. Detection of time-varying signals in event-related fMRI designs. *Neuroimage*. 2008 Nov 15;43(3):509-20. doi: 10.1016/j.neuroimage.2008.07.065. Epub 2008 Aug 16. PMID: 18775784; PMCID: PMC2654219.

Poldrack, R., Mumford, J., & Nichols, T. (2011). *Handbook of Functional MRI Data Analysis*. Cambridge: Cambridge University Press. doi:10.1017/CBO9780511895029

REVIEWER COMMENTS

Reviewer #1 (Remarks to the Author):

The authors have once again addressed my prior comments. The stated conclusions are supported by the data and I have no further suggestions.

Reviewer #3 (Remarks to the Author):

I appreciate the authors' efforts in revising their paper and addressing my last set of comments.

With regards to the gPPI analysis, I concur that the use of a single voxel could potentially introduce a greater degree of noise - however, in my opinion, it would then make sense to move towards a sphere approach or using the identified significant clusters of activity rather than entire ROIs, particularly given the identified circumscribed clusters of activity that are central to the study's conclusions. In any case, I'm not sure that the reported gPPI findings add any additional clarity to the interpretation of the main findings and the reported shift in hemispheric involvement is somewhat difficult to interpret in light of existing knowledge regarding detailed vs. gist-like representations.

Reviewer #5 (Remarks to the Author):

Summary:

The authors designed a clever "Mars Mission" paradigm whereby participants made judgements about different aspects of a multi-year mission while imagining themselves at various stages of the mission. I thought this was a nice way of inducing temporal judgments about the "past" and "future" within the confines of a ~1 hour experiment in the scanner. The paper reports that, relative to the present moment, representations of temporally distant (past or future) goals or events appear to be maintained by the anterior hippocampus, whereas representations of ongoing (current) goals or events are more localized to the posterior hippocampus. Overall this was a nicely executed study. The paradigm and analyses were appropriate, and the conclusions are justified.

Detailed comments:

I was asked to provide comments as a "replacement" reviewer, and as such I focused primarily on the comments and concerns raised by Reviewer #4. Relative to the other reviewers, Reviewer #4 was clearly more "skeptical" of the main findings and interpretations than the others. However, I am largely in agreement with the other reviewers and with the authors' responses. In summary, I think the authors have done a nice (and thorough) job of responding to prior comments. Reviewer #4 also raised concerns about potential alternative approaches and framings (e.g., of the introduction). While I appreciate that other approaches were "possible," I do not agree with Reviewer #4 that those other suggested approaches are inherently better than the current approaches, or that they are necessary for justifying the authors' conclusions or interpretations. Similarly, while other framings are of course possible and reasonable, I found the current framing compelling. I think the paper makes an important contribution to the field.

I have two relatively minor suggestions:

As Reviewer #4 points out, the "Remote > Current" comparison (Table 2, bottom row) yielded reliable responses in the right hippocampus. Therefore I would suggest slightly toning down the language about lateralization. Since the authors acknowledge that they did not have a priori hypotheses about the lateralization, I don't see the effect in right hippocampus as inconsistent with their broader

framing. I also recognize that there is certainly **more** of an effect in the left hippocampus. In Figure 2a, there seems to be an asymmetry between reaction times to distant future versus distant past goals. For example, the reaction times to goals from the most distant past (yellow curve, rightmost point) appear quite similar to the reaction times for the "current year" goals (middle of the panel, all curves). By eye, it also looks like the reaction times to future goals are generally higher than reaction times to equidistant past goals. If this asymmetry holds, the authors might consider renaming the figure caption to "Participants took longer to process temporally removed future goals than current goals," or some other title that acknowledges this asymmetry.

Dear editor and reviewers,

Thank you for the additional comments on the revision. We believe we were able to fully address the concerns, particularly the main concern of reviewer #3 regarding the supplementary analysis. We indicate the comments, followed by our responses and the text changes that resulted.

Reviewer #3 (Remarks to the Author):

I appreciate the authors' efforts in revising their paper and addressing my last set of comments. With regards to the gPPI analysis, I concur that the use of a single voxel could potentially introduce a greater degree of noise - however, in my opinion, it would then make sense to move towards a sphere approach or using the identified significant clusters of activity rather than entire ROIs, particularly given the identified circumscribed clusters of activity that are central to the study's conclusions. In any case, I'm not sure that the reported gPPI findings add any additional clarity to the interpretation of the main findings and the reported shift in hemispheric involvement is somewhat difficult to interpret in light of existing knowledge regarding detailed vs. gist-like representations.

Response: Thank you for your insightful comments on the gPPI analysis we reported in Supplementary Figure 6. As suggested, we performed the analyses on spherical ROIs from peak voxels with diameters of 9mm and 5mm. We did not find any significant results in our analyses and discovered that these spheres extended beyond the hippocampus's boundaries, potentially incorporating noise from surrounding areas. Shown below is the 5mm sphere in green and the 9mm sphere in orange.

9mm sphere

5mm sphere

As a result, we chose to remain with the previous anatomically defined masks of the anterior and posterior hippocampus, which provided a more precise representation of the regions of interest.

However, we recognize the concern about the difficulty of interpreting these findings in light of existing knowledge about detailed versus gist-like representations. According to the literature, hippocampal function is often lateralized, with the left hippocampus responsible for detailed, associative spatial long-term memory, and the right hippocampus for broader, spatial navigation tasks (e.g., Spiers et al., 2001). Our gPPI results show differential involvement of cortical areas with the anterior and posterior parts of the left hippocampi, suggesting a more complex and nuanced

interaction than previously thought. The observed shift in hemispheric involvement is consistent with the discrete-continuous hypothesis, which holds that the left and right hippocampi contribute to spatial memory and navigation in complementary ways (for a review Jordan et al., 2020). Our preliminary findings indicate that this complementary relationship could extend beyond the hippocampus to include broader cortical networks. Further, detailed and gist-like memory representations may not be strictly lateralized but rather involve integrated processing across both hemispheres. More broadly, these findings highlight the dynamic nature of brain networks during various types of memory processing. Instead of remaining static, these brain networks may adapt their connectivity, aiding in the understanding of memory mechanisms in both healthy and diseased states. This is in line with recent findings emphasizing lateralization's critical role in maintaining hippocampal function, which is frequently disrupted in neurological disorders (Nemati et al., 2023). We fully recognize that our interpretations are preliminary and should be approached with the scientific rigor and skepticism that new findings require.

Action taken: We have revised the caption of Supplementary Figure 6 to precisely define the regions in each hemisphere where we observed activation with our posterior and anterior seeds. We have also added the following text to supply readers with our interpretations in *Supplementary Figure 6*:

"We found that when goals were in the current moment, the left posterior hippocampus showed stronger association with regions in the right hemisphere, while the anterior part was primarily linked to regions in the left hemisphere. However, when goals were removed in time, both the left posterior and anterior hippocampus exhibit a greater association with regions in the right hemisphere. This pattern suggests a dynamic shift in hemispheric involvement based on the temporal context of the goals. These results align with the notion that different brain regions coordinate activity in response to specific cognitive demands and suggest that dynamic brain networks may be recruited during various types of memory processing (i.e. current versus remote goals). Further investigation is needed to understand the nuanced nature in which subregions of the hippocampus interact with bilateral cortical regions when processing temporal distance."

Reviewer #5 (Remarks to the Author):

Summary:

The authors designed a clever "Mars Mission" paradigm whereby participants made judgements about different aspects of a multi-year mission while imagining themselves at various stages of the mission. I thought this was a nice way of inducing temporal judgments about the "past" and "future" within the confines of a ~1 hour experiment in the scanner. The paper reports that, relative to the present moment, representations of temporally distant (past or future) goals or events appear to be maintained by the anterior hippocampus, whereas representations of ongoing (current) goals or events are more localized to the posterior hippocampus. Overall this was a nicely executed study. The paradigm and analyses were appropriate, and the conclusions are justified.

Detailed comments:

I was asked to provide comments as a "replacement" reviewer, and as such I focused primarily on the comments and concerns raised by Reviewer #4. Relative to the other reviewers, Reviewer #4 was clearly more "skeptical" of the main findings and interpretations than the others. However, I am largely in agreement with the other reviewers and with the authors' responses. In summary, I think

the authors have done a nice (and thorough) job of responding to prior comments. Reviewer #4 also raised concerns about potential alternative approaches and framings (e.g., of the introduction). While I appreciate that other approaches were possible, I do not agree with Reviewer #4 that those other suggested approaches are inherently better than the current approaches, or that they are necessary for justifying the authors conclusions or interpretations. Similarly, while other framings are of course possible and reasonable, I found the current framing compelling. I think the paper makes an important contribution to the field.

I have two relatively minor suggestions:

1) As Reviewer #4 points out, the “Remote > Current” comparison (Table 2, bottom row) yielded reliable responses in the right hippocampus. Therefore I would suggest slightly toning down the language about lateralization. Since the authors acknowledge that they did not have a priori hypotheses about the lateralization, I don’t see the effect in right hippocampus as inconsistent with their broader framing. I also recognize that there is certainly **more** of an effect in the left hippocampus.

Response: Thank you for acknowledging the need to highlight our bilateral activation.

Action taken: As suggested, we have re-named Table 2 to reflect the bilateral activation we observed as follows, *“Table 2: Coordinates of peak activation for temporally removed and current goals in the left and right hippocampus”* (line 280). To tone down the language about lateralization, we have added the following sentence to the summary of our results section (line 308), *“We observed this temporal dissociation to a lesser degree in the right hippocampus, which similarly showed stronger anterior activation for temporally removed past and future goals.”*

We have also adjusted our discussion to tone down the lateralization language by replacing the following sentence *“We found that goals were mapped based on their temporal distance in the left, but not the right hippocampus.”* (line 416) with the following sentence: *“We found that goals were mapped based on their temporal distance more robustly in the left hippocampus than in the right hippocampus.”*

2) In Figure 2a, there seems to be an asymmetry between reaction times to distant future versus distant past goals. For example, the reaction times to goals from the most distant past (yellow curve, rightmost point) appear quite similar to the reaction times for the “current year” goals (middle of the panel, all curves). By eye, it also looks like the reaction times to future goals are generally higher than reaction times to equidistant past goals. If this asymmetry holds, the authors might consider renaming the figure caption to Participants took longer to process temporally removed future goals than current goals, or some other title that acknowledges this asymmetry.

Response: Thank you for your comment on Figure 2(a). The two distant past conditions and two distant future conditions were not intended to be studied independently. Due to the design of the task, the 2 years into the past condition shown in Figure 2a had 10 trials and the 3 years into the past condition had 5 trials. These were combined into the single “distant past” category shown in Figure 2b so that the distant past condition had the same number of trials as the near past condition (1 year in the past = 15 trials). This was also the rationale for the combination of the distant future category. This Figure was designed to provide a detailed and precise account of how participants' reaction times changed as the temporal relevancies shifted. Figure 2 (a) helps readers understand the details of the participants' responses over time.

In Figure 2(b), we focused on comparing current goals to all temporally removed goals. This was consistent with our main objective, which was to investigate the overall pattern of reaction times across different temporal distances, without focusing on whether these goals were in the past or future. This approach is consistent with the primary focus of our research, as evidenced by our main behavioral findings and corresponding fMRI analyses.

We have decided to continue with the original title for coherence and simplicity. However, we recognize the significance of your comments. To address this, we performed additional *post hoc* analyses and found that distant past goals were processed more quickly than all temporally removed past and future goals when compared individually (distant future, near future, and near past). However, regarding the asymmetry, we did not observe a difference in the amount of time it took participants to process near past goals versus near future goals.

Action taken: We have added these results to Supplementary Table 2 and updated the text describing the table.

Contrast	Estimate	SE	df	t.ratio	P value	Significance
current vs. distant future	-0.227	0.023	2294	-10.05	<0.0001	***
current vs. near future	-0.24	0.021	2294	-11.62	<0.0001	***
current vs. near past	-0.28	0.021	2294	-13.54	<0.0001	***
current vs. distant past	-0.141	0.022	2294	-6.39	<0.0001	***
distant future vs. distant past	-0.086	0.029	2294	-2.95	0.026	**
near future vs. near past	0.039	0.023	2294	1.67	0.451	
near future vs. distant past	-0.098	0.025	2294	-3.81	0.001	***
near past vs distant past	-0.137	0.022	2294	-6.03	<0.0001	***

Supplementary Table 2. *Post hoc* reaction time comparisons for current versus temporally removed goals. We modeled participants log transformed reaction times against temporal condition, with levels for distant future, near future, current, near past, and distant past trials. Participants processed current goals more quickly than all temporally removed past and future goals when compared individually via *post hoc* contrasts. Participants processed distant past goals more quickly than all temporally removed past and future goals when compared individually via *post hoc* contrasts. Estimated marginal means and standard errors for the temporal conditions were as follows: distant future (1.34 ± 0.051), near future (1.35 ± 0.051), current (1.11 ± 0.050), near past (1.39 ± 0.051), distant

past (1.25 ± 0.049). The estimates above represent *post hoc* comparisons of the estimated marginal means for each condition. The *p* values were corrected for multiple comparisons using the Tukey method for comparing a family of 5 estimates. Degrees of freedom were approximated using the Satterthwaite method.

References:

Jordan J. T. (2020). The rodent hippocampus as a bilateral structure: A review of hemispheric lateralization. *Hippocampus*, 30(3), 278–292. <https://doi.org/10.1002/hipo.23188>

Seyed Saman Nemati, Leila Sadeghi, Gholamreza Dehghan, Nader Sheibani. (2023). Lateralization of the hippocampus: A review of molecular, functional, and physiological properties in health and disease. *Behavioural Brain Research*, 454, 114657. DOI: 10.1016/j.bbr.2023.114657.

REVIEWERS' COMMENTS

Reviewer #3 (Remarks to the Author):

I thank the authors for dealing with my last set of comments and have no further issues to raise at this point, which would improve the manuscript. I still think that the interpretation of the findings is, in some aspects, uncertain and somewhat speculative - at this stage, however, the data are the data and other readers can make their own judgment!

Reviewer #5 (Remarks to the Author):

The authors have done a nice job of addressing my prior comments. I have no further suggestions.